# Optimization of Ultrasound-Assisted Extraction of Polyphenols from *Ilex latifolia* Using Response Surface Methodology and Evaluation of Their Antioxidant Activity

**DOI:** 10.3390/molecules27133999

**Published:** 2022-06-22

**Authors:** Ying Chen, Xuqiang Sun, Lanting Fang, Xinxiu Jiang, Xuena Zhang, Zijun Ge, Rongbin Wang, Cunqin Wang

**Affiliations:** 1School of Pharmacy, Wannan Medical College, Wuhu 241002, China; 20190061@wnmc.edu.cn (Y.C.); 20200997@stu.wnmc.edu (X.S.); 18225928871@163.com (L.F.); 15055661846@163.com (X.J.); zxn01diligent@163.com (X.Z.); 13955231674@163.com (Z.G.); 2Anhui Provincial Engineering Research Center for Polysaccharide Drugs, Anhui Province Key Lab of Active Biological Macromolecules, Anhui Provincial Engineering Laboratory for Screening and Re-Evaluation of Active Compounds of Herbal Medicines in Southern Anhui, Wuhu 241002, China; 3Institute of Chinese Medicine Resources, Scientific Research Department, Anhui College of Traditional Chinese Medicine, Wuhu 241000, China

**Keywords:** *Ilex latifolia*, polyphenols, response surface methodology, antioxidant activity

## Abstract

The polyphenolic extract of *Ilex latifolia* (PEIL) exhibits a variety of biological activities. An evaluation of the parameters influencing the ultrasonic extraction process and the assessment of PEIL antioxidant activity are presented herein. Response surface methodology (RSM) was used to optimize the experimental conditions for the polyphenols ultrasonic-assisted extraction (UAE) from the leaves of *Ilex latifolia*. We identified the following optimal conditions of PEIL: ethanol concentration of 53%, extraction temperature of 60 °C, extraction time of 26 min and liquid–solid ratio of 60 mL/g. Using these parameters, the UAE had a yield of 35.77 ± 0.26 mg GAE/g, similar to the value we predicted using RSM (35.864 mg GAE/g). The antioxidant activity of PEIL was assessed in vitro, using various assays, as well as in vivo. We tested the effects of various doses of PEIL on D-galactose induced aging. Vitamin C (Vc) was used as positive control. After 21 days of administration, we measured superoxide dismutase (SOD) and glutathione peroxidase (GSH-Px) activities, malondialdehyde (MDA) levels in mouse serum and liver tissue. The results demonstrated that the PEIL exhibits potent radical scavenging activity against 1,1-diphenyl-2-picrythydrazyl (DPPH∙), 2,2′-azino-bis(3-ethylbenzothiazoline-6-sulfonic acid (ABTS^+^), and hydroxyl (∙OH) radicals. The serum concentrations of SOD and GSH-Px were higher, and MDA levels were lower, in the medium- and high-dose PEIL-treated groups than those in the aging group (*p* < 0.01), and the activity of MDA was lower than those of the model group (*p* < 0.01). The liver concentrations of SOD and GSH-Px were higher (*p* < 0.05), and MDA levels were lower, in the medium- and high-dose PEIL-treated groups than those in the aging control group (*p* < 0.01). These results suggest that optimizing the conditions of UAE using RSM could significantly increase the yield of PEIL extraction. PEIL possesses strong antioxidant activity and use as a medicine or functional food could be further investigated.

## 1. Introduction

Oxidative stress (OS) refers to the overproduction of greatly reactive molecules such as reactive oxygen species and reactive nitrogen species. Most of these molecules carry unpaired electrons and are known as free radicals. OS occurs when the body is subjected to various harmful stimuli that determine neutrophil inflammatory infiltration, increased protease secretion, and production of numerous oxidative intermediates, resulting in tissue damage [1]. It has a tremendous negative effect on the body, and is considered an underlying factor in aging and several chronic diseases such as diabetes, hypertension, atherosclerosis, and cancer [2,3].

Consuming various antioxidants is essential to overcome the harmful effects of free radicals and restore the normal balance between oxidants and antioxidants [4]. Synthetic antioxidants may have adverse effects on human body; therefore, people have been looking for new sources of natural compounds with antioxidant activity. Many edible vegetables, fruits, herbs, and woody plants are important sources of natural antioxidants, as they are rich in polyphenols, vitamins such as A, C, and E, and carotenoids [5,6]. Polyphenols are abundant secondary metabolites in plants [7]. The ortho-hydroxy moieties on their main phenolic structure are very easily oxidizable and have a strong affinity for free radicals such as ROS, thereby being effective scavengers [8]. Therefore, it is important to extract polyphenolic compounds from plants and to further use them as natural antioxidants for the prevention and treatment of various diseases induced by OS.

*Ilex latifolia* Thunb. is an evergreen tree belonging to the genus *Ilex* of the family Aquifoliaceae. It is widely distributed, yet mainly concentrated in Jiangsu, Zhejiang, Hubei, Anhui, and other places [9]. It has a variety of effects, such as clearing the head, dispersing heat, strengthening the stomach and eliminating accumulation, brightening the eyes, and benefiting thoughts. It is frequently used to treat mouth and tooth pain, red eyes, fever, and dysentery [10]. An analysis of the chemical constituents in its composition showed that *Ilex latifolia* mainly contains polyphenols, triterpenoids, triterpenoid saponins, and steroids [9,11,12]. Polyphenol monomers with high content in the leaves of *Ilex latifolia* mainly include chlorogenic acid, cryptochlorogenic acid, neochlorogenic acid, caffeic acid, rutin, and isochlorogenic acid A, B, and C [11]. Among them, chlorogenic acid has been deeply studied in the direction of antioxidant activity, and currently has significant value in nutritional drugs and food additives [13]. Other polyphenols with high content also have outstanding antioxidant activity [14,15], among which rutin and cryptochlorogenic acid can inhibit oxidative stress response [16,17]. Thus, research on the extraction process and antioxidant activity of the PEIL is of great interest.

In ultrasonic-assisted extraction (UAE), the ultrasonication-induced effects cavitation, vibration, crushing, and stirring contribute to the extraction process of the natural product components by disrupting the cell wall. This technique can shorten the extraction time and effectively improve efficiency [18]. It is an important tool for screening and optimizing various parameters in the extraction process. RSM was employed in this experiment. It uses a multiple quadratic regression equation to fit the functional relationship between factors and response values and selects the best process parameters through the regression analysis. It offers the advantages of a short test period, high accuracy of the regression equation, and the possibility to investigate the interactions between multiple factors [19].

Gallic acid was used as the standard in this study, and the content of PEIL was determined by the Folin–Ciocalteu method. Based on the results of single-factor experiments, the PEIL extraction conditions were optimized using RSM. Four main influencing factors were considered: ethanol concentration, extraction temperature, extraction time, and liquid–solid ratio. The free radical scavenging ability of the extract was measured in vitro, against DPPH∙, ABTS+, and ∙OH and in vivo. In vivo antioxidant assessment of PEIL included the determination of the activities of SOD and GSH-Px, as well as MDA levels in mouse serum and tissue. This study provides an optimized technique for *Ilex latifolia* polyphenols extraction and evaluates the antioxidant properties of the obtained extract.

## 2. Results

### 2.1. Single-Factor Experimental Analysis

In this study, the yield of PEIL was determined using the Folin–Ciocalteu method [20], gallic acid was used as a standard, and a standard curve was drawn (Figure 1). Four key parameters influencing UAE including ethanol concentration, extraction temperature, extraction time, and liquid–solid ratio, were selected for investigation. The effect of varying ethanol concentration on the yield of PEIL was studied. Ethanol concentrations ranged from 30% to 80%, when other extraction conditions were fixed as follows: extraction temperature 40 °C, extraction time 20 min, and liquid–solid ratio (30 mL/g). When increasing ethanol concentration, the extraction yield of PEIL first increased and then decreased (Figure 2a). Extraction with 50% ethanolic solution showed the highest yields among all ethanol concentrations tested. Polyphenol extraction increased in the initial stage of the process due to the increase of the polyphenols’ solubility in ethanol [21]. Polyphenol extraction ulteriorly decreased possibly due to the protein denaturation caused by the high concentration of ethanol. This change reduces the process of mass transfer, resulting in the decrease in polyphenol extraction [22]. Therefore, three ethanol concentrations, ranging from 40% to 60%, were selected for the optimization experiment.

The influence of the extraction temperature on the extraction yield of PEIL was investigated. We varied the temperature from 30 to 80 °C while keeping the other independent factors constant as follows: ethanol concentration at 50%, extraction time of 20 min, and liquid–solid ratio 30 mL/g. In Figure 2b, shows an increase in the extraction yield of PEIL at 30–60 °C, possibly because the increase in temperature, decreases the viscosity between polyphenols, enhancing the solute diffusion rate, and softens the cell wall. All these phenomena increase the extraction yield. At 60–70 °C, the extraction yield dropped due to the alteration of the polyphenols structure and their consequent inactivation. At 70–80 °C, the yield of polyphenols increased again, possibly due to the enhancement of some heat-sensitive polyphenols extraction [23]. Consequently, a temperature range of 50–70 °C is a suitable temperature for the PEIL extraction.

The influence of the different extraction time on the PEIL extraction yield is illustrated in Figure 2c. The other independent variables were kept constant: ethanol concentration of 50%, extraction temperature of 60 °C, and liquid–solid ratio of 30 mL/g. The PEIL extraction yield increased rapidly when the extraction time increased from 10 to 25 min, possibly due to the acceleration of the fragmentation or deformation of tissue cells. This promotes the dissolution of polyphenols, increasing the concentration of phenolic compounds extracted. The PEIL extraction yield declined markedly after 25 min, maybe owing to polyphenol structural degradation or oxidation [24]. Therefore, an extraction time ranging from 20 to 30 min was selected for the optimization process.

To study the effect of the liquid–solid ratio on the PEIL extraction yield, different liquid–solid ratios ranging from 20 to 70 mL/g were examined, while keeping the other extraction conditions constant as follows: ethanol concentration of 50%, extraction temperature 60 °C, and extraction time 25 min. As shown in Figure 2d, the PEIL extraction yield reached the maximum value of 35.27 mg GAE/g at a liquid–solid ratio of 60 mL/g. At higher ratios, the extraction yield decreased. A larger liquid–solid ratio indicates a greater concentration gradient between the solute and the solvent at the surface of the raw material. This determines a quicker diffusion of polyphenols, and an enhancement of the extraction yield [25]. However, when the liquid–solid ratio was further increased, the ultrasonic energy attached to the unit volume decreased, resulting in reduction of the extraction yield [26]. Therefore, a liquid–solid ratio ranging from 50–70 mL/g was considered optimal.

### 2.2. Model Fitting

A four-factor and three-level Box-Behnken design (BBD) of RSM was employed for the analysis of the single factor experiment results. The four analyzed factors were ethanol concentration (40%, 50%, 60%), extraction temperature (50 °C, 60 °C, 70 °C), extraction time (20 min, 25 min, 30 min) and liquid–solid ratio (50 mL/g, 60 mL/g, 70 mL/g). The design comprised 29 detection points: 24 analysis points and 5 zero estimation errors. Responses ranged as 30.91 to 36.1 mg GAE/g. This indicates that the extraction conditions clearly affected the yield of the PEIL.

After all the experimental combinations of different extraction variables were considered, the results were fitted into Equation (1) with coded factors for the PEIL extraction yield (*Y*). Using a quadratic model, the following Equation was obtained:*Y* = 35.75 + 0.51A + 0.27B + 0.094C − 0.24D − 0.75AB − 0.087AC + 0.090AD + 0.057BC + 0.043BD − 0.21CD − 0.87A^2^ − 2.32B^2^ − 0.23C^2^ − 0.55D^2^(1)
where A is ethanol concentration (%), B is the extraction temperature (°C), C is the extraction time (min), and D is the liquid–solid ratio (mL/g).

A summary of the analysis of variance (ANOVA) results for the BBD-based model terms is given in Table 1. High *F*-values (17.49) and small *p*-values (*p* < 0.001) suggest that the developed regression models are significant. The quality of the model fit was assessed using the determination (*R^2^*) and adjusted determination coefficients (*R^2^_adj_*). Their values were 0.9459 and 0.8918, respectively, indicating a high degree of correlation between the experimental and predicted values. Furthermore, the low coefficient of variation (CV) (1.24%) and high precision (15.461) indicate a very high degree of precision and good reliability of the experimental values. The linear coefficients (A and B), quadratic term coefficients (A^2^, B^2^, and D^2^), and cross-product coefficients (AB) were significant (*p* < 0.05) and influenced extraction yield, while the coefficients of the other terms were not (*p* > 0.05). The *F* values confirmed that the order of factors influencing the PEIL yield extraction was ethanol concentration> extraction temperature> liquid–solid ratio> extraction time, and the order of the interaction effects was AB > CD > AD > AC > BC > BD.

### 2.3. Analysis of Response Surface

Three-dimensional (3D) response surface plots based on Equation (1) illustrate the relationship between the extraction parameters and the PEIL extraction yield. Under the condition that other experimental factors were fixed, the influence of interaction terms on the yield was investigated, and the response surface analysis graph was used to evaluate the pairwise interaction of experimental factors on the PEIL yield [27]. The steepness of the response surface indicates the degree in which paired experimental variables influence extraction yield [19]. As depicted in Figure 3a, the slope of the response surface changed greatly by increasing the ethanol concentration and extraction temperature, suggesting that these variables have a great impact on PEIL extraction yield. High extraction time and liquid–solid ratio were associated with a reduced slope change of the response surface, indicating that they have little impact on the PEIL extraction yield, as illustrated in Figure 3b–e, and f. The significance of the interaction of each factor on the PEIL extraction yield is consistent with the results of ANOVA presented in Table 1.

To determine the optimal extraction conditions, an analysis of the response surface was performed using Design Expert software. The PEIL extraction parameters optimized by RSM were as follows: ethanol concentration of 52.58%, extraction temperature of 60.183 °C, extraction time of 26.399 min, and liquid–solid ratio of 57.444 mL/g. Under these conditions, the estimated PEIL amount to be obtained was 35.864 mg GAE/g. Thereafter, an extraction was triply carried out with ethanol concentration 53% and liquid–solid ratio of 60 mL/g at 60 °C for 26 min. The amount of PEIL obtained was 35.77 ± 0.26 mg GAE/g, which was close to the theoretical value, further validating our analysis method.

### 2.4. PEIL Antioxidant Activity

#### 2.4.1. In Vitro Assessment of Antioxidant Activity

##### DPPH Radical Scavenging Activity

The DPPH methanolic solution was purple and presented a strong absorption at 517 nm. When a free radical scavenger interacted with DPPH, the color of the solution became lighter, and the absorbance value at 517 nm decreased [28]. PEIL scavenging ability against DPPH free radical is shown in Figure 4a. Vc was used as a positive control. PEIL’s scavenging ability and that of the control were positively correlated with sample concentration. When the scavenging ratios of PEIL and Vc to DPPH were basically stable, their concentrations and scavenging ratios were 1.6, 0.02 mg/mL, 89.82% and 96.06%, respectively. Although PEIL had a lower scavenging activity than Vc, it was higher than that reported in other studies [29]. These results indicate that PEIL has a powerful ability to transfer electron or hydrogen atoms to DPPH, thus possessing antioxidant effect.

##### ABTS Radical Scavenging Activity

ABTS assay is important for measuring the antioxidant capacity of plant extracts. The free radical scavenging ability of extracts against ABTS reflects their ability to provide electrons to the oxidant, neutralizing it. The antioxidant activity of natural products was assessed by measuring the decolorization reaction of ABTS. Stable free-radical positive ions have blue-green chromophores. The presence of an antioxidant reduces the intensity of the coloration [30]. PEIL radical scavenging activity against ABTS is illustrated in Figure 4b. This activity is concentration-dependent. When the concentration of PEIL was low, its ABTS scavenging ability was weak. When the concentration reached 1.6 mg/mL, the scavenging ratio increased significantly, and at a concentration of 3.2 mg/mL, the ratio increased to 82.71%. For Vc concentrations of 0.8 mg/mL, the ABTS scavenging rate reached 92.51%, which was different from the DPPH method. The results indicate that PEIL has an evident antioxidant activity.

##### Hydroxyl Radical Scavenging Activity

Hydroxyl free radicals are very reactive oxygen radicals that can alter almost all cellular constituents. No enzyme in the human body withstand its effect, phenomenon leading to serious cellular damage [31]. Thus, counteracting hydroxyl radical is important for the protection of living systems. The PEIL radical-scavenging activity against hydroxyl radical is presented in Figure 4c. When the concentration of PEIL was in the range of 0.05~0.40 mg/mL. PEIL scavenging ability against hydroxyl radicals remained basically constant, up to 79.83% and was higher than that of the Vc. That suggests a low concentration of PEIL could have a strong scavenging effect against hydroxyl radicals. Increasing the concentration improved the scavenging rate of PEIL, with the highest value being 91.21%. Although the high-concentration PEIL scavenging ability against hydroxyl radicals was slightly lower than that of Vc, it still possessed strong antioxidant activity.

#### 2.4.2. Antioxidant Activity In Vivo

##### PEIL Effects on Serum SOD and GSH-Px Activities and MDA Content, in Mice 

Antioxidant function can regulate the balance of oxidation/antioxidant system, scavenge free radicals, and reduce the damage of free radicals to the body. The body mainly relies on the activity of antioxidant enzymes such as SOD, GSH-Px, and CAT to produce antioxidant effects [32]. MDA is the main product of lipid peroxidation, and its content reflects the degree of damage to the membrane system [33]. The effects of different treatments on mouse serum SOD, GSH-Px activities, and MDA content are shown in Table 2. Compared with group II, group I presented significantly higher serum SOD and GSH-Px activities and lower MDA content (*p* < 0.01), indicating that oxidative stress was successfully induced. Compared with group II, group IV presented increased SOD (*p* > 0.05) and GSH-Px (*p* < 0.01) activities and lower MDA content (*p* > 0.05). The activities of SOD and GSH-Px were significantly higher in groups V and VI than in group II (*p* < 0.01), and the MDA content was significantly lower (*p* < 0.01). The highest-administered PEIL dose demonstrated stronger antioxidant activities than Vc.

##### Effects of PEIL on Liver SOD and GSH-Px Activities and MDA Content, in Mice

The effects of different treatments on liver SOD and GSH-Px activities, and MDA content in mice are shown in Table 3. Compared with group II, group I had significantly higher liver SOD and GSH-Px activities (*p* < 0.01), and significantly lower MDA levels (*p* < 0.01), which further supports the utilized aging model. Compared with group II, group IV presented increased SOD (*p* < 0.05) and GSH-Px (*p* > 0.05) activities and lower MDA content (*p* < 0.01).The activities of SOD and GSH-Px were significantly higher and the MDA content was significantly lower in groups V and VI than in the aging control group (*p* < 0.01). The highest-administered PEIL dose had the most intense antioxidant effect.

## 3. Discussion

Conventional extraction methods include immersion, decoction, percolation, and reflux extraction methods. However, owing to disadvantages such as poor selectivity, long duration, an associated risk of product impurification, and easy degradation of active ingredients, these methods can negatively affect the pharmacological activity and the stability of these traditional medicinal products [34]. UAE can significantly improve the extraction rate of polyphenols and flavonoids from plants [35,36]. This method is based on the “cavitation effect”, thus promoting the dissolution of soluble components. We used UAE to increase the efficiency of the PEIL extraction process. The experimental analysis method is an important for quickly optimizing extraction conditions. Compared with the frequently used orthogonal experimental method, RSM has apparent advantages. The functional relationship between the fitting factors of the multiple quadratic regression equation and the response value was evaluated by using the experimental data, and the optimal process parameters were finally determined. The analysis of the regression equation not only solves the multivariable problem but also refines the number of experiments needed, ensuring a reasonable experimental design [37]. Therefore, in this experiment, RSM was used to optimize the PEIL extraction process, based on the results of single-factor experiments. ANOVA demonstrated that the ethanol concentration and extraction temperature had an extremely significant effect on the PEIL extraction yield (*p* < 0.01; *p* < 0.05). The CV is the ratio of the standard deviation to the average value, and its magnitude is related to the repeatability of the experiment. Lower CV values indicate less variation in the mean and better repeatability of the experiment [38]. When the CV was less than 10%, the model accuracy was high [39]. The CV value of this experiment was 1.24%, which showed that the RSM results were consistent with the experimental results and the model was reliable. Free radicals are harmful compounds that are produced during oxidative reaction in the body. High levels of free radicals cause irreversible oxidative damage to the body, leading to chronic diseases and aging [40]. Natural and synthetic phenols are widely used as free radical inhibitors in medicine, chemical industry, food industry and other fields [41]. The DPPH method is a fast and feasible method for estimating the free radical scavenging ability of antioxidants [42]. The ABTS free radical scavenging method can be performed to measure the antioxidant activity of multicomponent mixtures and to calculate the overall antioxidant activity of biological samples [43]. Hydroxyl free radicals are the most oxidizing type of free radicals in nature, and they undergo chemical reactions with most of the components in the human body. Thus, they play a crucial role in the progression of various diseases [44]. Therefore, in this study, we used these three in vitro methods to evaluate PEIL antioxidant activity. The results showed that the scavenging rates of PEIL (3.20 mg/mL) against DPPH and ABTS free radicals were 96.06% and 92.51%, respectively, which is higher than those of other traditional Chinese medicine extracts [29,45]. PEIL scavenging rate against hydroxyl radicals at low concentrations (0.05–0.40 mg/mL) was significantly higher than that of Vc, suggesting that PEIL can be used as a natural antioxidant for preventing various diseases caused by oxidative stress. Antioxidant mechanisms can regulate the balance of pro-oxidation/antioxidants, eliminate free radicals, and reduce the damage caused by free radicals to the body. These mechanisms mainly depend on the activities of SOD, GSH-Px, CAT, and other antioxidant enzymes. SOD, GSH-Px activities and MDA level are important markers of oxidative stress and are commonly used to evaluate the degree of aging [46,47]. Animal aging models are usually obtained by administering D-galactose to rodents [48]. In this experiment, the D-galactose-induced aging mouse model was used to assess the PEIL antioxidant activity in vivo, by measuring SOD and GSH-Px activities and MDA level in the mouse serum and liver. The study showed that after a week of D-galactose intraperitoneal injection, the mice in the aging model group presented symptoms such as slow movement and dull hair color, whereas the mice in the positive group and PEIL groups presented no such changes. This indicates that PEIL could have an effective therapeutic effect against D-galactose-induced oxidative stress. Serum and liver SOD and GSH-Px activities were significantly increased in PEIL-treated mice compared with the aging control group. These changes were significant for groups V and VI (*p* < 0.01). The serum and liver MDA levels were significantly lower in group V and VI mice than t in the group II mice (*p* < 0.01). The serum MDA levels in group IV mice were slightly lower than that in group II mice (*p* > 0.05). In the serum of aging mice, the antioxidant activity of groups V and VI was better than that of group III, although it was not apparent in the liver. Therefore, PEIL could serve as effective antioxidant agents.

The results of this study showed that the use of RSM to optimize the conditions of UAE could significantly improve PEIL extraction yield. Therefore, our work provided a powerful approach for simultaneous determination of multi-components of other traditional Chinese medicines. PEIL has good antioxidant activity in vivo and in vitro and could be further developed into natural antioxidant drugs to prevent aging, atherosclerosis, and cancer caused by oxidative stress. 

## 4. Materials and Methods

### 4.1. Materials

*Ilex latifolia* plant samples used in this study were collected from Nanjing, Jiangsu Province, China, in July 2019. The specimen was authenticated by Prof. Rongbin Wang, and a voucher specimen was deposited at the College of Pharmacy, Wannan Medical College, Wuhu, China.

Kunming mice, male, 20–25 g, clean grade, approval number: SCXK (Yu) 2020-0005, were purchased from Henan Skebes Biotechnology Co., Ltd. (Anyang, China).

DPPH was purchased from the TCI (Shanghai, China) Chemical Industry Development Co., Ltd. (Shanghai, China). (ABTS^+^·), Folin–Ciocalteu phenol reagent, 6% H_2_O_2_, K_2_(SO4)_2_, CMC-Na, Vc, D-(+)-galactose were purchased from Shanghai Aladdin Biochemical Technology Co., Ltd. (Shanghai, China). Gallic acid was obtained from Shanghai McLean Biochemical Technology Co., Ltd. (Shanghai, China). SOD, GSH-Px, MDA, total protein kit was purchased from Nanjing Jiangcheng Bioengineering Institute. (Nanjing, China). The rest of the chemical reagents were analytical grade reagents, and the water was ultrapure water. The instruments used in this study were: KQ-500DE CNC Ultrasonic Cleaner (Kunshan Ultrasonic Instrument Co., Ltd. Jiangsu, China), Aquapro AWL-1001-M Ultrapure Water Machine (Chongqing Yiyang Enterprise Development Co., Ltd., Chongqing, China), Ultrospec 7000 UV-Vis Spectrophotometer (Cytiva, Marlborough, MA, USA) and MULTISKAN G0 Microplate Reader (Thermo Fisher Scientific, Waltham, MA, USA).

### 4.2. PEIL Extraction

In this experiment, 0.1 g powder obtained from the leaves of *Ilex latifolia* was accurately weighed and poured into 15 mL cuvettes with stoppers. PEIL were obtained by ultrasonic under different experimental conditions. The ultrasonic power was 250 W, and the frequency was 40 kHz. The extract was filtered using a 0.45 μm microporous membrane and a PEIL solution was obtained.

### 4.3. Determination of Polyphenol Content

The PEIL total phenolic content was evaluated by spectrophotometry with the Folin–Ciocalteu method, some modifications were made [20]. Briefly, 1 mL of each solution of the different PEILs (diluted 20 times) was measured into test tubes, 1 mL of Folin–Ciocalteu reagent was added to each, stirred evenly, and kept for 2 min. Next, 6 mL of 7.5% Na_2_CO_3_ solution was added and 6 mL of ultrapure water were added, and the solution was mixed thoroughly. After incubation in dark for 1 h, the absorbance was measured at 760 nm. Gallic acid solutions of various well-known concentrations were submitted to the same procedure and used to plot a calibration curve. The PEIL total phenolic content was calculated as previously described [49] with a slight change:(2)$$\mathrm{Yield}\;\mathrm{of}\;\mathrm{PEIL}\;(\mathrm{mg}\;\mathrm{GAE}/g)=\frac{c\times n\times V}{m\times 1000}$$
where *c* is the mass concentration of PEIL (μg/mL), *n* is the dilution factor, *V* is the sample volume (mL), and *m* is the sample mass (g).

### 4.4. Response Surface Methodology

#### 4.4.1. Variables Selection

The extraction yield of plants is affected by numerous factors, including extraction time, extraction temperature, solvent type, particle size and liquid–solid ratio [50]. In this experiment, four factors, namely ethanol concentration, extraction temperature, extraction time, and liquid–solid ratio, which greatly influence the effect of the polyphenols extraction rate from medicinal materials, were selected as variables [51]. As mentioned previously, the particle size affects the extraction yield [52]. Therefore, 100 mesh size was selected in our experiment. To determine a suitable range of factors, single-factor analysis was conducted using the following variables: ethanol concentration of 30%, 40%, 50%, 60%, 70%, and 80%; extraction temperature of 30, 40, 50, 60, 70, and 80 °C; extraction time of 5, 10, 15, 20, 25, and 30 min; liquid–solid ratios of 20, 30, 40, 50, 60, and 70 mL/g. One level on both sides was selected to determine the three levels of these four factors. The values of these variables associated with the highest polyphenol extraction amount were subsequently selected for the RSM design. 

#### 4.4.2. BBD for Extraction Optimization

Factor analysis by RSM is used in many different fields of research to screen out the best extracting technology, determine the effective parameters, and improve the performance [53]. Ethanol concentration (A, %), extraction temperature (B, °C) extraction time (C, min), and liquid–solid ratio (D, mL/g) were selected as independent variables. A four-factor, three-level BBD was constructed. The PEIL extraction yield obtained from triplicate experiments were designated as the response–dependent value. Table 4 presents the variables and their levels, with both coded and actual values. The analysis scheme and results of the experiments are provided in Table 5. According to experimental data of BBD, regression analysis was performed, and the second-order polynomial model was performed as follows:(3)$$Y=\beta_{0}+\sum_{i=1}^{4}(\beta_{i}x_{i})+\sum_{i=1}^{3}\sum_{j=i+1}^{4}(\beta_{ij}x_{i}x_{j})+\sum_{i=1}^{4}(\beta_{ii}x_{i}^{2})$$
where *Y* is the predicted response; *β*_0_ is a constant, *β_i_* is linear coefficient, *β_ii_* is a quadratic coefficient, and *β_ij_* is an interaction coefficient. *x_i_* and *x_j_* are independent variables.

### 4.5. Antioxidant Activity In Vitro

#### 4.5.1. DPPH Radical Scavenging Activity

PEIL scavenging activity against DPPH radicals was measured as previously reported [54] with minor modifications. Briefly, 1.0 mL of PEIL solution (0.05 to 3.20 mg/mL) was thoroughly mixed with 6.0 mL freshly prepared DPPH methanolic solution (0.1 mmol/L). The mixture was incubated at 25 °C in the dark for 65 min, and then was measured at 517 nm. Vc was taken as a control. The scavenging activity was calculated according to the following equation:(4)$$\mathrm{Scavenging}\;\mathrm{activity}(\%)=(1-\frac{A_{1}-A_{2}}{A_{0}})\;\times \;100$$
where A_0_ is the the absorbance of the blank DPPH solution, A_1_ the absorbance of a mixture solution containing both the sample and DPPH, and A_2_ is the absorbance of the sample solution without DPPH. 

#### 4.5.2. ABTS Radical Scavenging Activity

The scavenging activity of PEIL against ABTS radical cation (ABTS^+^) was evaluated according to the method described by Cui Tong et al. with minor modifications [55]. A 20 μL aliquot of PEIL solution (0.05 to 3.20 mg/mL) was mixed with 4 mL of ABTS^+^ solution and left for 6 min at 25 °C. Absorbance was then measured at 734 nm. ABTS radical scavenging activity was calculated according to using Equation (4), where A_0_ is the absorbance of the blank ABTS solution, A_1_ is the absorbance of a mixture solution containing both the sample and ABTS, and A_2_ is the absorbance of the sample solution without ABTS. Vc was regarded as the control. 

#### 4.5.3. Hydroxyl Radical Scavenging Assay

The scavenging activity of PEIL against the hydroxyl radical was evaluated on basis of the method described by Li et al. [56]. Briefly, 2 mL of PEIL solution (0.05 to 3.20 mg/mL) was mixed 2.0 mL of 6 mmol/L solutions of both FeSO_4_ and H_2_O_2_. The mixtures were kept at 25 °C for 15 min. Then, 2.0 mL of freshly prepared 6 mmol/L salicylic acid ethanolic (SAE) solution was added and mixed. After incubating for 6 min at 37 °C, the absorbance was measured at 510 nm. Equation (4) was also used to calculate the PEIL scavenging activity, where A_0_ is the absorbance of the blank SAE solution, A_1_ is the absorbance of a mixture solution containing both the sample and SAE, and A_2_ is the absorbance of the sample solution without SAE. Vc was taken as the control.

### 4.6. In Vivo Evaluation of the Antioxidant Activity

#### 4.6.1. Animals

Kunming mice of age 6–7 weeks, weighing 20–25 g, were raised in the Animal Laboratory of Wannan Medical College. All animals were housed in a comfortable environment with 12 h light-dark cycle, 20–25 °C temperature, and 40–60% humidity. Animals had ad libitum access to rodent food and water. This study was affirmed by the Animal Welfare and Ethics Committee of Wannan Medical College (protocol number: LLSC-2021-192). 

#### 4.6.2. Experimental Design

After one week of acclimation, 48 mice were randomly divided into six groups (*n* = 8). This experiment adopted the mode of drug administration during modeling. First, groups II–VI were intraperitoneally administered D-galactose (i.p., 400 mg/kg/d). Group I mice were administered with normal saline (i.e., 0.1 mL/10 g). Then, the extracts and drug were administered after an hour: group I: normal saline (control group); group II: normal saline (aging control mice); group III: 100 mg/kg/day Vc (positive control receiving standard antioxidant drug); groups IV, V, and VI: 36.87, 110.61, and 331.79 mg/kg/d of PEIL, respectively. The extracts and drug were administered to the animals once daily via gavage for 21 days. 

#### 4.6.3. Biochemical Assay

After 21 days, the animals were submitted to fasting for 12 h, anesthetized with ether, and blood was collected through retro-orbital plexus. Mice were sacrificed by cervical dislocation. The livers were collected, cleaned with normal saline, and dried. Pre-cooled normal saline was added to the liver in a ratio of 1:9 (m/V), homogenized with a homogenizer, centrifuged at 2000 r/min for 15 min, and the supernatant was collected for the determination of tissue biochemical indices. The blood samples were centrifuged at 4000 rpm for 10 min, and then the serum was gathered and frozen at −80 °C for further testing. The serum and liver levels of SOD, GSH-Px, and MDA were measured using a commercial ELISA kit based on the manufacturer’s instructions. 

### 4.7. Statistical Analysis

All the measurements were repeated three times. BBD was planned using Design Expert software, version 10.0.1. The results of SOD, GSH-Px, and MDA levels in aging groups are expressed as means ± SD. Data were evaluated with SPSS 19.0 (SPSS Inc., Chicago, IL, USA). Statistical significance of differences between groups was assessed using one-way ANOVA, followed by LSD test. The significance level was set at *p* < 0.05. Graphs were drawn using the GraphPad Prism software (version 9.0, San Diego, CA, USA).

## 5. Conclusions

In this present study, UAE was used to extract polyphenols from *Ilex latifolia* leaves, and RSM was successfully employed for the optimization of extraction conditions. The ethanol concentration was the most influential variable of the PEIL extraction process, followed by extraction temperature, whereas the extraction time and liquid–solid ratio did not significantly influenced the process. Through the analysis of the 3D response surface plots, we identified the following optimal conditions: ethanol concentration of 53%, extraction temperature of 60 °C, extraction time of 26 min, liquid–solid ratio of 60 mL/g. Using these conditions, we obtained a PEIL solution with a total phenolic content similar to the estimated one. Thus, our method of analysis was validated. The results indicate that PEIL had strong scavenging activity against DPPH, ABTS, and OH radicals. Furthermore, the PEIL effect on aging was studied, using a D-galactose-induced aging mouse model. PEIL administration enhanced the activity of antioxidant enzymes and decreased lipid peroxidation. This study confirmed the natural product, PEIL, has antioxidant effects. This study focused on the polyphenol extract of *Ilex latifolia*, but failed to locate the monomer component. Thus, further studies need to purify the PEIL and investigate the antioxidant mechanisms of the polyphenols from *Ilex latifolia*.

## Figures and Tables

**Figure 1 molecules-27-03999-f001:**
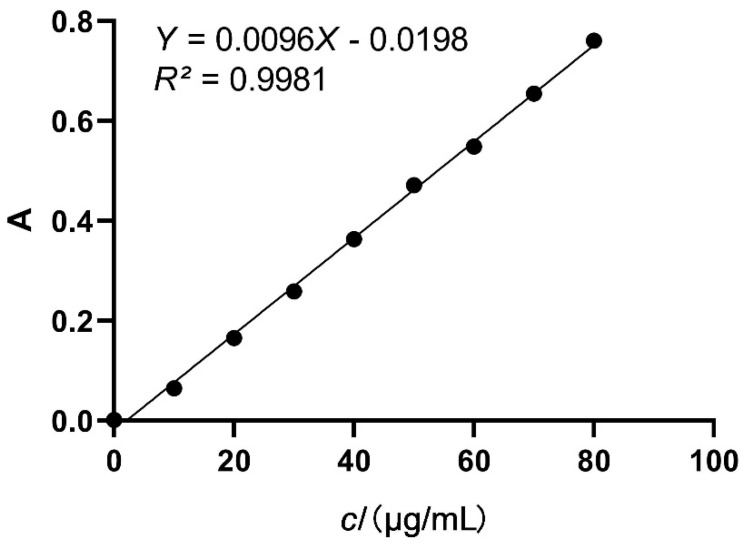
The calibration curve of standard gallic acid.

**Figure 2 molecules-27-03999-f002:**
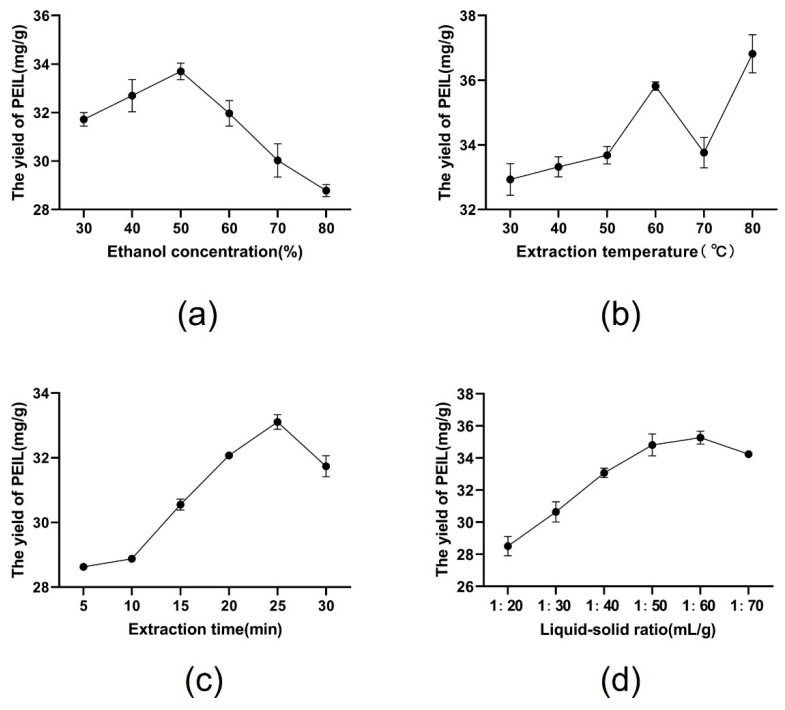
Effects of different extraction parameters on the yield of PEIL: (**a**) ethanol concentration, (%); (**b**) extraction temperature, (°C); (**c**) extraction time, (min); (**d**) liquid–solid ratio, (mL/g).

**Figure 3 molecules-27-03999-f003:**
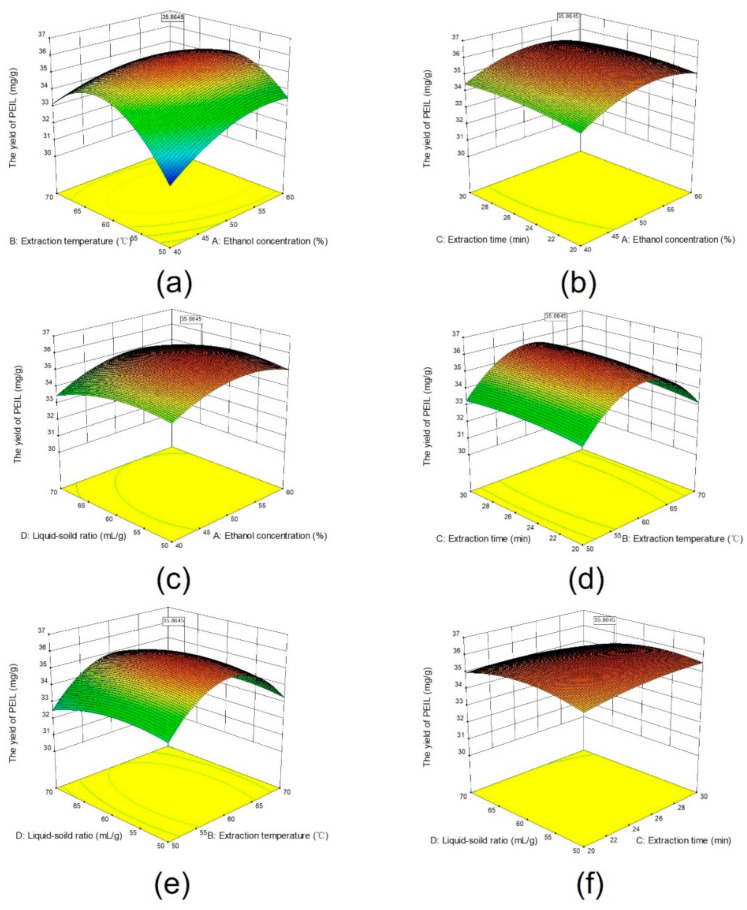
Surface plots of interactive effects of ethanol concentration(A), extraction temperature (B), extraction time (C), and liquid–solid ratio (D) on the yield of PEIL: (**a**) (A) and (B); (**b**) (A) and (C); (**c**) (A) and (D); (**d**) (B) and (C); (**e**) (B) and (D); (**f**) (C) and (D).

**Figure 4 molecules-27-03999-f004:**
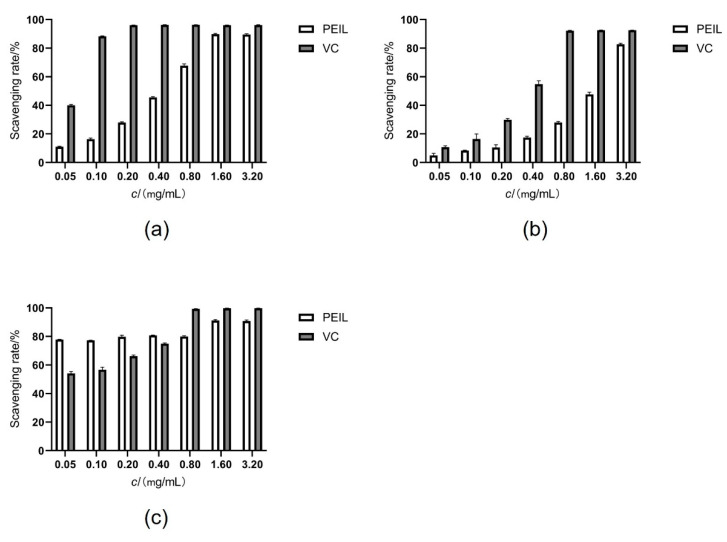
Antioxidant activities of PEIL and Vc with different methods in vitro. (**a**) Scavenging activity for DPPH·; (**b**) ABTS^+^-scavenging activity; (**c**) ·OH-scavenging assay.

**Table 1 molecules-27-03999-t001:** Analysis of variance (ANOVA) of the quadratic model and lack of fit.

Source of Variance	Sum of Squares	df	Mean Square	*F*-Value	*p*-Value	Signature
Model	43.89	14	3.13	17.49	<0.0001	**
A	3.09	1	3.09	17.24	0.0010	**
B	0.90	1	0.90	5.00	0.0421	*
C	0.11	1	0.11	0.59	0.4538	
D	0.72	1	0.72	4.02	0.0647	
AB	2.22	1	2.22	12.39	0.0034	**
AC	0.031	1	0.031	0.17	0.6856	
AD	0.032	1	0.032	0.18	0.6772	
BC	0.013	1	0.013	0.074	0.7899	
BD	7.225 × 10^−3^	1	7.225 × 10^−3^	0.040	0.8438	
CD	0.18	1	0.18	1.01	0.3325	
A^2^	4.87	1	4.87	27.17	0.0001	**
B^2^	34.77	1	34.77	193.99	<0.0001	**
C^2^	0.33	1	0.33	1.86	0.1947	
D^2^	1.95	1	1.95	10.86	0.0053	**
Residual	2.51	14	0.18			
Lack of fit	2.28	10	0.23	4.0	0.0970	
Pure error	0.23	4	0.057			
Cor total	46.39	28				

Note: ** means extremely significant when *p* < 0.01, * means significant when *p* < 0.05.

**Table 2 molecules-27-03999-t002:** Serum oxidative stress parameters of mice treated with PEIL.

Group	SOD/(U·mg^−1^)	GSH-Px/(U·mg^−1^)	MDA/(nmol·mg^−1^)
Normal control (Group I)	268.25 ± 5.22 **	175.00 ± 29.65 **	8.19 ± 0.523 **
Model control (Group II)	254.05 ± 5.26	137.01 ± 22.97	10.76 ± 0.5
Positive control (Group III)	277.38 ± 4.62 **	169.92 ± 23.85 **	9.25 ± 0.40 **
PEIL-L (Group IV)	259.18 ± 10.29	166.45 ± 13.25 **	9.76 ± 2.12
PEIL-M (Group V)	274.55 ± 3.92 **	171.20 ± 12.10 **	7.86 ± 0.63 **
PEIL-H (Group VI)	282.59 ± 1.27 **	183.96 ± 10.59 **	8.13 ± 1.03 **

Note: ** *p* < 0.01: comparing with model.

**Table 3 molecules-27-03999-t003:** Liver oxidative stress parameters of mice treated with PEIL.

Group	SOD/(U/mg)	GSH-Px/(U/mg)	MDA/(nmol/mg)
Normal control (Group I)	84.33 ± 2.59 **	198.27 ± 24.54 **	1.51 ± 0.06 **
Model control (Group II)	76.02 ± 1.65	151.21 ± 8.5	1.95 ± 0.08
Positive control (Group III)	83.60 ± 4.55 *	222.53 ± 28.38 *	1.26 ± 0.13 **
PEIL-L (Group IV)	83.77 ± 2.57 **	175.70 ± 36.26	1.47 ± 0.02 **
PEIL-M (Group V)	93.80 ± 4.87 **	182.28 ± 25.18 *	1.50 ± 0.13 **
PEIL-H (Group VI)	87.69 ± 1.75 **	213.46 ± 22.76 **	1.46 ± 0.11 **

Note: * *p* < 0.05: comparing with model; ** *p* < 0.01: comparing with model.

**Table 4 molecules-27-03999-t004:** The natural and coded values of independent variables used in Box–Behnken design (BBD).

Factor	Code	Level		
		−1	0	1
Ethanol concentration/%	A	40	50	60
Extraction temperature/°C	B	50	60	70
Extraction time/min	C	20	25	30
Liquid–solid ratio/(mL/g)	D	50	60	70

**Table 5 molecules-27-03999-t005:** Box–Behnken design (BBD) for the independent variables and corresponding response values.

Run	A	B	C	D	The Yield of PEIL (mg GAE/g)	
					Actual Value	Predicted Value
1	−1	−1	0	0	30.97	31.03
2	1	−1	0	0	34.2	33.55
3	−1	1	0	0	32.6	33.07
4	1	1	0	0	32.85	32.59
5	0	0	−1	−1	34.56	34.906
6	0	0	1	−1	35.4	35.514
7	0	0	−1	1	35.15	34.846
8	0	0	1	1	35.14	34.614
9	−1	0	0	−1	34.39	34.15
10	1	0	0	−1	34.87	34.99
11	−1	0	0	1	33.25	33.49
12	1	0	0	1	34.09	34.69
13	0	−1	−1	0	32.58	32.893
14	0	1	−1	0	33.22	33.319
15	0	−1	1	0	32.71	32.967
16	0	1	1	0	33.58	33.621
17	−1	0	−1	0	34.4	33.959
18	1	0	−1	0	35.22	35.153
19	−1	0	1	0	34.48	34.321
20	1	0	1	0	34.95	35.167
21	0	−1	0	−1	33	32.893
22	0	1	0	−1	33.66	33.347
23	0	−1	0	1	32.24	32.327
24	0	1	0	1	33.07	32.953
25	0	0	0	0	35.73	35.75
26	0	0	0	0	36.1	35.75
27	0	0	0	0	35.6	35.75
28	0	0	0	0	35.48	35.75
29	0	0	0	0	35.85	35.75

## Data Availability

The data presented in this study are available on request from the corresponding author.

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
