# Peer review of "Optimization of Ultrasound-Assisted Extraction of Polyphenols from Ilex latifolia Using Response Surface Methodology and Evaluation of Their Antioxidant Activity"

_molecules, 2022, doi:10.3390/molecules27133999_

Round 1
Reviewer 1 Report
The paper can be published with minor revision and may recommendations are:
- some minor correction of English language (i.e line 399 more recommended is Folin reagent)
- some paragraphs must be rephrased. Statements such as "The traditional extraction methods of traditional Chinese medicine include immersion method, decoction method, percolation method, reflux extraction method, etc." are not really correct. For plants all over the world are used these extraction methods, not only in Chinese medicine.
- conclusion chapter must be improved .It is very similar with the abstract
Reviewer 2 Report
The manuscript entitled “Optimization of Ultrasound-Assisted Extraction of polyphenols from Ilex latifolia Using Response Surface Methodology and Evaluation of Their Antioxidant Activities” mainly dealt with the rise in polyphenols. The similarity index is 40%, which should be addressed.
1) Figure 1 seems to be the calibration curve of standard gallic acid. The regression line should pass through the origin. I am speculating on the recorded absorbance values with various concentrations of gallic acid. Please review the related literature or repeat the experiment.
2) Usually, we mention Folin–Ciocalteu method (not Folin phenol method) for the measurement of total phenolic content. The determination of total polyphenolic content is not convincing. Where is the use of gallic acid standard? TPC should be expressed in mg GAE/g.
3) Please write down the correct formula of chemicals used for example H2O2, K2(SO4)2, where numbers should be subscript. There are several such issues to be addressed in the manuscript.
4) Also, you need to report antioxidant activities with IC50 values. The relative percentiles are not sufficient.
5) Total flavones content is also responsible for the increased antioxidant activity.
6) The authors should mention the influence of ultrasonic amplitude.
7) Please do mention possible phenolic acids responsible for antioxidant activity through the literature review.